# Hyporheic hydraulic geometry: Conceptualizing relationships among hyporheic exchange, storage, and water age

**Geoffrey C. Poole**[1,2]*, **S. Kathleen Fogg**[1], **Scott J. O'Daniel**[3], **Byron E. Amerson**[1], **Ann Marie Reinhold**[1,2], **Samuel P. Carlson**[1], **Elizabeth J. Mohr**[1], **Hayley C. Oakland**[1]

**1** Department of Land Resources and Environmental Sciences, Montana State University, Bozeman, Montana, United States of America, **2** Montana Institute on Ecosystems, Montana State University, Bozeman, Montana, United States of America, **3** Confederated Tribes of the Umatilla Indian Reservation, Pendleton, Oregon, United States of America

* gpoole@montana.edu

## Abstract

Hyporheic exchange is now widely acknowledged as a key driver of ecosystem processes in many streams. Yet stream ecologists have been slow to adopt nuanced hydrologic frameworks developed and applied by engineers and hydrologists to describe the relationship between water storage, water age, and water balance in finite hydrosystems such as hyporheic zones. Here, in the context of hyporheic hydrology, we summarize a well-established mathematical framework useful for describing hyporheic hydrology, while also applying the framework heuristically to visualize the relationships between water age, rates of hyporheic exchange, and water volume within hyporheic zones. Building on this heuristic application, we discuss how improved accuracy in the conceptualization of hyporheic exchange can yield a deeper understanding of the role of the hyporheic zone in stream ecosystems. Although the equations presented here have been well-described for decades, our aim is to make the mathematical basis as accessible as possible and to encourage broader understanding among aquatic ecologists of the implications of tailed age distributions commonly observed in water discharged from and stored within hyporheic zones. Our quantitative description of "hyporheic hydraulic geometry," associated visualizations, and discussion offer a nuanced and realistic understanding of hyporheic hydrology to aid in considering hyporheic exchange in the context of river and stream ecosystem science and management.

## Introduction

The magnitude and spatial extent of hyporheic exchange—the continuous bidirectional exchange of water between the stream channel and underlying sediments—is determined largely by the interactions between stream flows, channel form, and hydraulic properties of the streambed and alluvial aquifer sediments [1]. In fine-grained streambeds, the hyporheic zone may be restricted to the top few hundredths of a meter of the streambed [2, 3]. In rivers and

**Data Availability Statement:** The data used to produce Figs 4–8 have been uploaded to a public

repository, DOI: https://doi.org/10.5061/dryad.m905qfv2q.

**Funding:** This research was funded by US DOE - Bonneville Power Administration Project # 2007-252-00 (SJO; https://www.bpa.gov/Projects/Initiatives/Pages/Columbia-River-Treaty.aspx), and from the National Science Foundation (BIO-DEB Award 1945941) and the USDA National Institute of Food and Agriculture (Hatch Project 1015745) (GCP; https://nifa.usda.gov/program/hatch-act-1887). The funders had no role in study design, data collection and analysis, decision to publish, or preparation of the manuscript.

**Competing interests:** The authors have declared that no competing interest exist.

montane streams where alluvial sediments consist primarily of coarse sands, pebbles, gravels, and/or cobbles, expansive coarse-grained alluvial aquifers may have hyporheic zones ranging from tenths to tens of meters thick and extending 1's to 1000's of meters laterally from the stream channel [4–8]. Regardless of extent, hyporheic zones are important drivers of stream biogeochemistry [9–11]. Expansive hyporheic zones may also exchange enough water with the channel to influence surface water temperature [12, 13]. At coarser spatial scales, the cumulative effect of hyporheic exchange can govern whole-ecosystem processes such as respiration and nutrient uptake in stream networks [14].

Hydrologists and stream ecologists have been expounding and refining conceptual models of hyporheic exchange for decades [15–19]. As a result of such work, most aquatic scientists accept that hyporheic exchange is an important driver of hydrologic and ecosystem processes in streams [20, 21], and some understand that hyporheic flow paths are nested hierarchically—that the majority of hyporheic exchange traverses relatively short flow paths in the hyporheic zone. Yet few scientists outside of hydrologists who study and model the hyporheic zone consider the inter-dependence among hyporheic hydrologic variables. Just as surface water exiting a stream reach has a distribution of transit time through the reach, emerging hyporheic water has a "water age distribution" describing the distribution of time that hyporheic discharge has spent within the hyporheic zone. Similarly, just as stream flow, water volume, and hydrologic residence time are inter-dependent in a stream reach, the rate of hyporheic exchange, magnitude of hyporheic water storage, and hyporheic water age distribution are inexorably linked. When considered alongside the shape of the water age distribution of hyporheic discharge (e.g., derived from tracer experiments), hydrologic inter-dependencies of hyporheic exchange rate, water volume, and water age can be used to infer important aspects of hyporheic hydrology in surprising detail.

The concept of interdependence among hydrologic variables in porous media hydrosystems is not new; the mathematics necessary to describe the relationships among flow rate, storage volume, and water age within steady-state porous media chemical reactors were published in a classic Chemical Engineering paper by Danckwerts [22]. In fact, these equations are included in basic Chemical Engineering textbooks that discuss the analysis, modeling, or design of chemical reactors [23]. Similar concepts have also been used to simulate transient storage (including hyporheic exchange) in streams [24, 25]. More recently, many of these same concepts have been extended to address non-steady state hydrosystems with arbitrarily complex water age distributions arising from transient (dynamic across time) rates of water flow [26–28]. The resulting "ranked StorAge Selection", or rSAS framework has been applied to describe dynamic hydrology in both watersheds [29] and hyporheic zones [30].

Our goal in compiling this paper is to encourage consideration of water age distributions by aquatic scientists who may currently employ less rigorous conceptualizations of hyporheic hydrology. Although we acknowledge that transient hydrology (e.g., flood spates) plays a critical role in hyporheic dynamics and therefore stream ecosystems, we focus on steady-state assumptions in this paper, leaving aside the dynamic case. We believe starting with steady-state assumptions of hyporheic hydrology is appropriate to illustrate the ecological implications of asymmetrical, heavily "tailed" water age distributions typically observed in hyporheic zones, and we choose to present and apply modifications of the steady-state equations presented by Danckwerts [22]. In doing so, we trade the generality and flexibility of the dynamic rSAS approach in favor of the simplicity and relative transparency of steady-state assumptions. We believe that consideration of steady-state hydrology provides a useful initial case for incorporating improved hydrologic realism into conceptual models of hyporheic exchange, and provides a necessary foundation for subsequent exploration of non-steady-state conditions such as those inherent in the rSAS framework.

## Water age distributions

Despite the fact that water exiting any hydrosystem is derived from water stored within the hydrosystem, the age distribution of exiting water is rarely equal to the age distribution of stored water. To understand this concept in the context of a hyporheic zone, consider the hyporheic zone as a collection of flow paths of different lengths. Each flow path begins at the channel, traverses some distance of the hyporheic zone, and re-enters the channel. The age distribution of hyporheic discharge represents water ages *at the end* of the hyporheic flow paths, while the age distribution of hyporheic water storage represents the water ages *along the length* of the flow paths. Thus, the paradox of different age distributions for hyporheic discharge and hyporheic storage is resolved by recognizing that exiting water is not a random sample of stored water.

Confusingly, the published terms used to describe the water age distribution of hydrosystem discharge vs. that of hydrosystem storage have become somewhat confounded. Specifically, one common convention refers to the water age distribution of hyporheic *discharge* as the hyporheic "residence time distribution," e.g., [25, 31, 32]. Another convention uses "residence time distribution" to refer to the age distribution of water *stored* within a hydrosystem, while "transit time distribution" describes the water age distribution of hydrosystem *discharge* [27, 28, 30]. For the sake of clarity, we leave aside names for the two age distributions and choose to distinguish them with notation: $AD_d$ (age distribution of hyporheic discharge) and $AD_s$ (age distribution of hyporheic water storage). Use of such notation has the advantage of reinforcing that $AD_d$ and $AD_s$ describe the same metric (water age) as applied to different aspects of a hydrosystem (water discharge vs. internal water storage).

## Methods

### Conceptual model

Stream beds and alluvial aquifers host a mixture of waters sourced from the channel, catchment soils, deeper aquifers, and precipitation. As a strategic simplification that helps illuminate the dynamics of hyporheic exchange, our conceptual model of hyporheic hydrology considers only subsurface water that originates from the channel and that will ultimately return to the channel. We refer to this water as "hyporheic water" and largely leave aside the term "hyporheic zone", except when used in the most general sense. Such hyporheic water comprises the majority of water found in the streambed, and often throughout coarse-grained alluvial aquifers [33].

We use the symbol $\tau$ to represent water age—the time elapsed since a parcel of hyporheic water left the stream channel and entered the alluvial aquifer. So defined, $\tau$ might vary from a fraction of a second to a year or more [34], depending on the size of the alluvial aquifer and the temporal scale of interest. $\tau_0$ and $\tau_n$ represent minimum and maximum water ages of interest, values which may or may not be zero and $\infty$, respectively, depending on the application of our conceptual model.

We adopt the aquifer-centric reference frame of hydrogeologists wherein water that enters the alluvial aquifer from the channel ("aquifer recharge"; $q_\downarrow$) is represented using a positive number and water returning to the channel from the aquifer ("aquifer discharge"; $q_\uparrow$) is represented as a negative number. We assume steady state conditions ($q_\downarrow = -q_\uparrow$) typical of hyporheic hydrology when river discharge is stable. Our conceptual model (and the equations presented below) can be applied to a length, area, or volume of water, so long as the dimension of hyporheic water storage ($s$), $q_\uparrow$, and $q_\downarrow$ are consistent. Specifically, depending on the length dimensions chosen, $q_\downarrow$ and $q_\uparrow$ can describe a vertical water flux ([L][T$^{-1}$]), a rate at which the cross-

or longitudinal-section area of hyporheic water exchanges with the cross- or longitudinal-section area of the channel ($[L^2][T^{-1}]$), or a rate at which the volume of hyporheic water exchanges with the volume of water in a stream reach ($[L^3][T^{-1}]$). Correspondingly, $s$ would be represented by a thickness of water beneath the wetted channel ($[L]$), a cross- or longitudinal-section area of hyporheic water ($[L^2]$), or a volume ($[L^3]$) of water stored beneath a channel reach. Therefore, we use the notation $[L^x][T^{-1}]$ to represent the dimensions of $q_\downarrow$ and $q_\uparrow$, and $[L^x]$ to represent the dimensions of $s$, where $x \in \{1, 2, 3\}$.

Generally, conventional conceptual models of hyporheic exchange consider the hyporheic zone to be a single unit. To visualize the relationships among $q_\downarrow$, $s$, the $AD_d$, and the $AD_s$, we subdivide the hyporheic zone into multiple transient storage zones (TSZs; Fig 1). Each TSZ ($i$) is defined by a maximum water age ($\tau_i$); the minimum water age of each TSZ is equal to $\tau_{i-1}$. Therefore, the ranges of water age across the TSZs are serial and contiguous. Importantly, our use of the notation $\tau_{i-1}$ to denote the minimum water age of $TSZ_i$, $\tau_i$ to denote the maximum water age of $TSZ_i$, and $n$ to denote the number of TSZs delineated in the hyporheic zone means that the symbol $\tau_0$ refers to both the minimum hyporheic water age of interest and the minimum hyporheic water age of $TSZ_1$ (which are numerically equivalent). Similarly, $\tau_n$ refers to both the maximum hyporheic water age of interest and the maximum hyporheic water age of $TSZ_n$ (values that are, again, numerically equivalent).

Several important characteristic of such TSZs are worth noting, and arise from the fact that TSZs are delineated using water age:

- All hyporheic recharge must enter the aquifer via $TSZ_1$ because recharge water has $\tau = \tau_0$.

- Because hyporheic discharge ranges in age from $\tau_0$ to $\tau_n$, every TSZ discharges water to the channel.

- By definition, when the age of hyporheic water exceeds $\tau_i$, the water becomes part of $TSZ_{i+1}$; over time, then, hyporheic water not discharged to the channel inhabits the TSZs serially, in order.

- TSZs are delineated by time (water age). Yet because hyporheic water often moves along flowpaths as it ages, we discuss water as moving across space (flowing) between TSZs. As a result, TSZs are delineated spatially by isochronal surfaces—surfaces within the aquifer defined by equal $\tau$ (Fig 2). For instance, the isochronal surface representing $\tau = \tau_i$ is the spatial boundary that delineates $TSZ_i$ from $TSZ_{i+1}$.

- Fig 1 displays each TSZ as contiguous, yet TSZs can be disaggregated (e.g., red TSZ in Fig 2); all water aged $\tau_{i-1} < \tau \leq \tau_i$ is part of $TSZ_i$ regardless of location within the aquifer.

By visualizing the patterns of water storage within, exchange among, and discharge from such a collection of TSZs arrayed across water ages, we can create quantitative graphical depictions that reveal fundamental relationships among hyporheic water age, water storage, and water flow in hyporheic zones.

## Quantifying hyporheic hydraulic geometry

Any $TSZ_i$ is fully characterised by six metrics: maximum water age ($\tau_i$), minimum water age ($\tau_{i-1}$), flow rate from the prior TSZ ($q_{\downarrow \tau_{i-1}}$), hyporheic flow rate to the next TSZ ($q_{\downarrow \tau_i}$), discharge to the channel ($q_{\uparrow \tau_{i-1}, \tau_i}$), and associated water storage $s_{\tau_{i-1}, \tau_i}$ (Fig 1). To describe the equations necessary to estimate each of these metrics, we adopt the conventional Chemical Engineering notation [23] ultimately derived from [22]. Specifically, we use three functions: the exit age

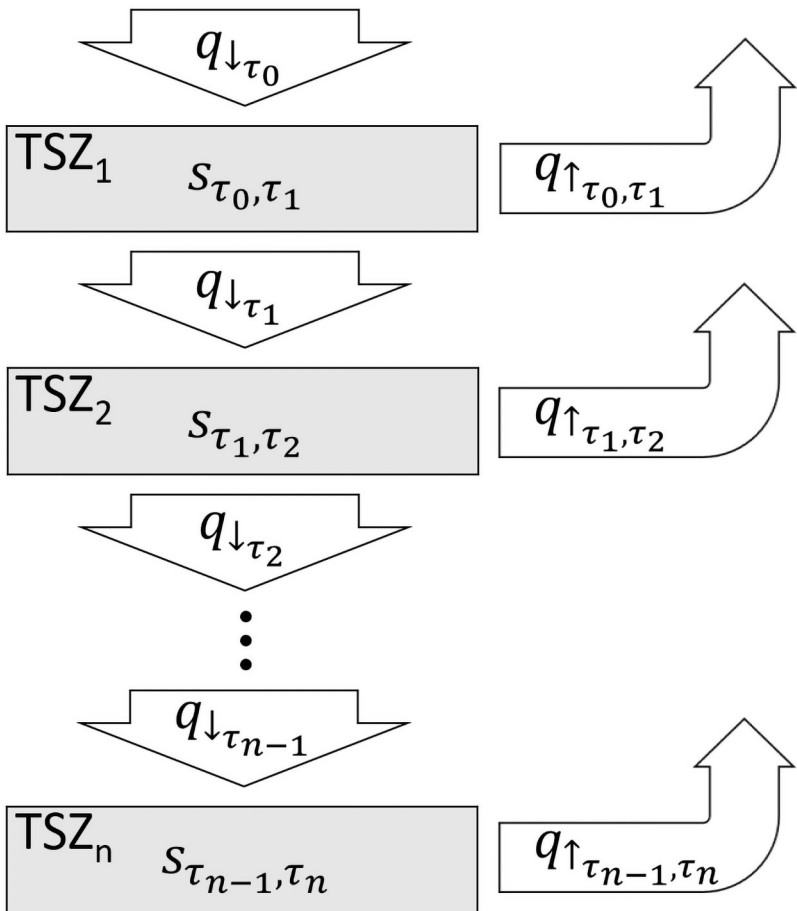

**Fig 1. Simple graphical representation of transient storage zone hydrology within a hyporheic zone.** Grey boxes represent transient storage zones (TSZs) with associated hyporheic water storage ($s$) with water age between $\tau_{i-1}$ and $\tau_i$. White arrows represent water recharging the aquifer ($q_{\downarrow\tau_0}$), flow between TSZs ($q_{\downarrow\tau_i}$) or water discharging from the aquifer ($q_{\uparrow\tau_{i-1},\tau_i}$).

density function ($E(\tau)$), the washout function ($W(\tau)$), and the internal age density function ($I(\tau)$).

$E(\tau)$ is a probability density function (PDF) with dimensions [$T^{-1}$]. Given $\tau^*$, a particular water age of interest, $E(\tau^*)$ returns the probability density that hyporheic water will return to the channel with a water age equal to $\tau^*$. Thus, $E(\tau)$ is a PDF representing the age distribution of hyporheic discharge ($AD_d$).

As with any probability density function, the area under the curve prescribed by $E(\tau)$ must be unity (so that the probability of hyporheic recharge returning to the channel at some time between $\tau_0$ and $\tau_n$ is 1.0):

$$\int_{\tau_0}^{\tau_n} E(\tau) \, d\tau = 1. \tag{1}$$

If we chose a function, $f(\tau)$ [dimensionless] to represent the desired shape of the $AD_d$ (and thus the shape of $E(\tau)$), we can create a similarly shaped PDF by dividing values returned by $f$

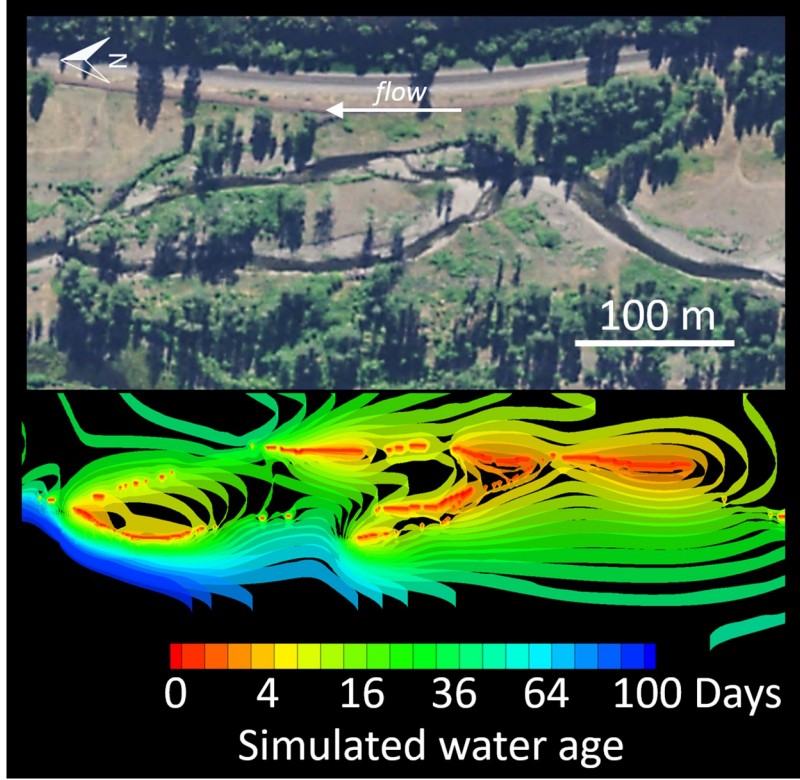

**Fig 2. Visualization of the non-contiguous nature of TSZs within floodplain aquifers.** Isochronal surfaces ("ribbons" in lower panel) demarcate TSZ's within floodplain alluvial aquifers. Upper panel shows an aerial photo of the floodplain surface. Visualization was created using simulation results derived from an application of the HydroGeoSphere model [35] to the Meacham Creek (Oregon, USA) floodplain restoration conducted by the Confederated Tribes of the Umatilla Indian Reservation (Byron Amerson, Unpublished). Aerial imagery from the National Agriculture Imagery Program [36].

($\tau^*$) by the integral of (area under) $f(\tau)$:

$$E(\tau^*) = \frac{1}{\int_{\tau_0}^{\tau_n} f(\tau) \, d\tau} f(\tau^*) \quad \tau_0 \leq \tau^* \leq \tau_n, \tag{2}$$

(The constraint $\tau_0 \leq \tau^* \leq \tau_n$ carries through the rest of the equations in this paper. For brevity, we omit the constraints from the remaining equations.) Eq 2 can be restated as:

$$E(\tau^*) = k f(\tau^*), \tag{3}$$

where $k$ [T$^{-1}$] is the reciprocal integral or "normalizing constant" that converts $f(\tau)$ to a PDF.

Any definite integral of $E(\tau)$ (i.e., from $\tau_a$ to $\tau_b$, where $\tau_a < \tau_b$) yields the probability [dimensionless] that hyporheic water will return to the channel with a water age $\tau_a \leq \tau \leq \tau_b$. Considered another way, the finite integral of $E(\tau)$ is the fraction of $q_\downarrow$ that will return to the channel with $\tau_a \leq \tau \leq \tau_b$:

$$q_{\uparrow \tau_a, \tau_b} = -q_\downarrow \int_{\tau_a}^{\tau_b} E(\tau) \, d\tau. \tag{4}$$

Thus, Eq 4 can be used to calculate $q_{\uparrow \tau_{i-1}, \tau_i}$ (Fig 1) for each TSZ. Remembering that the integral

of $E(\tau)$ from $\tau_0$ to $\tau_n$ is unity, Eq 4 also shows that $q_{\uparrow\tau_0,\tau_n} = q_\uparrow = -q_\downarrow$, as would be expected under steady state flow.

The washout function, $W(\tau)$, is the complementary cumulative distribution of $E(\tau)$, which is the integral of $E(\tau)$ from any $\tau^*$ to $\tau_n$:

$$W(\tau^*) = \int_{\tau^*}^{\tau_n} E(\tau)\ d\tau. \qquad (5)$$

$W(\tau)$ describes the probability [dimensionless] that hyporheic water will be discharged to the channel with an age greater than the specified value of $\tau^*$. Put another way, $W(\tau^*)$ determines the fraction of $q_\downarrow$ that remains in the alluvial aquifer at water age $\tau^*$. Therefore, $q_{\downarrow\tau}$ in Fig 1 can be calculated as:

$$q_{\downarrow\tau^*} = q_\downarrow\ W(\tau^*). \qquad (6)$$

Because $W(\tau^*)$ returns the fraction of $q_\downarrow$ remaining in the aquifer at water age $\tau^*$, $W(\tau)$ provides the correct shape for a PDF describing the distribution of hyporheic water age, known as the "internal age density function" ($I(\tau)$). Therefore:

$$I(\tau^*) = \frac{1}{\int_{\tau_0}^{\tau_n} W(\tau)\ d\tau}\ W(\tau^*). \qquad (7)$$

Because $I(\tau)$ is a PDF representing the age distribution of hyporheic water in the alluvial aquifer, the integral of $I(\tau)$ from $\tau_a$ to $\tau_b$ yields the fraction of $s$ (hyporheic water stored in the alluvial aquifer) having a water age $\tau_a \leq \tau \leq \tau_b$.

$$s_{\tau_a,\tau_b} = s\int_{\tau_a}^{\tau_b} I(\tau)\ d\tau \qquad (8)$$

Alternatively, we can think of water storage as the accumulation of flow over time. Since the integration of $W(\tau)$ represents accumulation of fractional remaining flow in the aquifer, we can also calculate the fraction of $s$ having a water age $\tau_a \leq \tau \leq \tau_b$ as the product of the rate of exchange and the finite integral of $W(\tau)$:

$$s_{\tau_a,\tau_b} = q_\downarrow\int_{\tau_a}^{\tau_b} W(\tau)\ d\tau \qquad (9)$$

Thus, Eq 9 can be used to calculate $s_{\tau_{i-1},\tau_i}$ (Fig 1) for each TSZ.

Eqs 4, 6 and 9 provide a means of quantifying our entire conceptual model (Fig 1) for any number ($n$) of TSZs defined in the hyporheic zone. Because $W(\tau)$ is derived from $E(\tau)$ and, in turn, $E(\tau)$ arises from $f(\tau)$, the only requirements to quantify our conceptual model are: 1) a choice of $f(\tau)$, which describes the desired shape of the $AD_d$; 2) a choice of $\tau_i$ for each desired TSZ (including $\tau_0$ and $\tau_n$); and 3) an estimate of $q_\downarrow$. Because values of $\tau_i$ are chosen to define each $TSZ_i$, only $q_\downarrow$ and $f(\tau)$ are unknown. Rearranging Eq 9 reveals that $q_\downarrow$ can be estimated by:

$$q_\downarrow = \frac{s}{\int_{\tau_0}^{\tau_n} W(\tau)\ d\tau}. \qquad (10)$$

Thus, by assuming a shape for $f(\tau)$, the only value required to describe the hydraulic geometry (Fig 1) of the hyporheic zone is $s$, the thickness, cross-sectional area, or volume of hyporheic water beneath a stream.

## Visualizing hyporheic hydraulic geometry

In some cases, simple field observations provide an estimate of $s$. For instance, bedrock-confined hyporheic zones common to montane systems often permeate the entire alluvial aquifer [33]. In these systems, values of $s$ can be approximated as the product of various aquifer dimensions (length, width, and/or depth, depending on the dimensions of $s$) and aquifer porosity. In other systems, values for $s$ may be more difficult to ascertain. Regardless, in the absence of an estimate of $s$ for a particular system, hydraulic geometry for a representative unit (RU) of the aquifer can be visualized.

We define an RU as an idealized hyporheic zone with unit volume ($s = 1$, $[L^x]$) which has, by definition, the same $AD_d$ and $AD_s$ as any larger hyporheic zone it represents. Usefully, $q_\downarrow$ scales linearly with $s$ (Eq 10) and the hydrologic metrics of any TSZ scale linearly with $q_\downarrow$ (Eqs 4, 6 and 9). Therefore, $s$ is a factor that converts $q_{\downarrow\tau_i}$, $q_{\uparrow\tau_{i-1},\tau_i}$, or $s_{\tau_{i-1},\tau_i}$ for an RU to the same value for the associated hyporheic zone. Further, ratios among $q_{\downarrow\tau_i}$, $q_{\uparrow\tau_{i-1},\tau_i}$, and $s_{\tau_{i-1},\tau_i}$ are identical for an RU and the associated hyporheic zone.

**Specifying $f(\tau)$.** Both a power-law [6, 25, 37] and exponential function [38] are commonly used to represent the $AD_d$ of hyporheic exchange in streams. In S1 Appendix, we present solutions for $E(\tau)$, $W(\tau)$, and $\int_{\tau_a}^{\tau_b} W(\tau)$ for both a power-law and exponential representation of an $AD_d$. Importantly, either representation yields a heavily tailed distribution where the median age of emerging water is often substantially lower than the mean; i.e., most hyporheic water is discharged back to the channel with an age somewhat younger than the mean age, while a small portion of the water exits with an age far greater than the mean. Thus, the choice of either a power law or exponential function yields a sufficiently skewed distribution for the purpose of conceptualizing and visualizing important aspects of hyporheic hydrology. Below, we illustrate hyporheic hydrology using a power law with a negative exponent to represent the shape of the $AD_d$:

$$E(\tau) \propto \tau^{-\alpha}, \tag{11}$$

although an appropriately parameterized exponential distribution would yield substantively similar results given our purpose.

Because such a power law approaches infinity as $\tau$ approaches zero, Expression 11 implies infinite hyporheic exchange rates at vanishingly small $\tau$. Application of Expression 11 is therefore eased by the choice of a suitably small minimum water age of interest ($\tau_0$) that is greater than zero. Additionally, hyporheic zones are not infinite flow systems. Therefore, we also set a maximum water age of interest ($\tau_n$) less than infinity. Values of $\tau_0 = 60$ s and $\tau_n = 3.1536 \times 10^7$ s (1 y) provided reasonable values to approximate and visualize the hydraulic geometry of a hyporheic zone similar to systems we study, e.g., the Nyack Floodplain of the Middle Fork Flathead River, Montana, USA [6, 39] and the main-stem Umatilla River, Oregon, USA [6, 33]. Thus, for our purposes here, the definition of "hyporheic water" excludes water that enters the stream bed but returns to the channel with $\tau < 60$ s, as well as water "lost" to any deeper groundwater flow system (i.e., with $\tau > 1$ y). In establishing $\tau_0$ and $\tau_n$, we are not arguing that the excluded water ages have no hydrologic or ecological importance. Rather, we chose values of $\tau_0$ and $\tau_n$ that are inclusive of water ages typical of hyporheic water in the expansive alluvial aquifers we study [39]. Graphically, selection of $\tau_0$ and $\tau_n$ results in a PDF that describes the shaded region in Fig 3. The function describing the height of the shaded region for any $\tau$ is:

$$f(\tau) = \tau^{-\alpha} - \tau_n^{-\alpha}. \tag{12}$$

Thus, Eq 12 provides the basis for $E(\tau)$ (Eq 2) in our application.

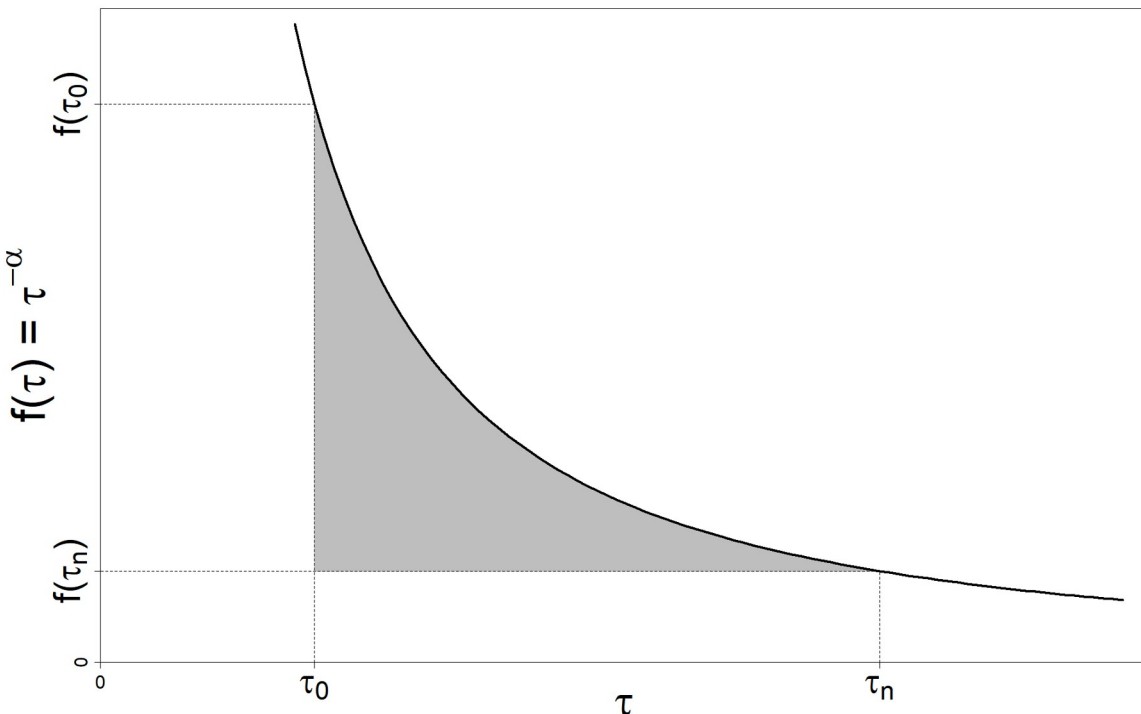

**Fig 3. Graphical representation of a probability density function proportional to a power-law.** The area (shaded) is used to determine a probability density function (PDF) defined by $\tau_0$ to $\tau_n$, assuming the PDF is proportional to a power law.

To visualize the hydraulic geometry of a hyporheic zone, we calculated hyporheic geometry metrics for an RU. Specifically, we subdivided the RU into 50 TSZs, individually denoted as $TSZ_i$. Values of $\tau_i$ (maximum water age in each $TSZ_i$) were determined such that hyporheic water storage was equal across TSZs (i.e., $s_i = 0.02$ m$^x$ for each of the 50 TSZs). Note that, by definition, notation referencing characteristics of each TSZ is synonymous with previously used notation; specifically $q_{\downarrow i}$ equates to $q_{\downarrow \tau_i}$, $q_{\uparrow i}$ equates to $q_{\uparrow \tau_{i-1}, \tau_i}$, and $s_i$ equates to $s_{\tau_{i-1}, \tau_i}$.

We surveyed the literature for empirical observations of $\alpha$ from tracer release experiments, finding that $\alpha$ ranges from approximately 1.3 to 1.9 (Table 1). To consider hyporheic zones across the observed range of $\alpha$, we calculated values of $q_{\downarrow i}$ and $q_{\uparrow i}$ for each $TSZ_i$ using four different values of $\alpha$: $\alpha \in \{1.3, 1.5, 1.7, 1.9\}$.

For each value of $\alpha$, we plotted cumulative hyporheic discharge across water ages, marking $\tau_i$ for each TSZ on the plot (Fig 4). Thus, the x-axis range between adjacent marks on each curve represents the range of water age associated with individual TSZs. The y-axis range between adjacent marks on each curve represents the the amount of water discharged from each TSZ to the channel.

We also summarize the relationships between $\tau_i$, $s_i$, and $q_{\uparrow i}$ for the hyporheic RU in quantitative depictions associated with each value of $\alpha$ (Fig 5). The depictions use a color map to show the age distribution of hyporheic water stored within an RU of a hyporheic zone. Superimposed on the color map is the volume of water that is discharged from each of the 50 hyporheic TSZs over a period of one hour. Whether for a single TSZ or for the entire hyporheic zone, the ratio of the colored area to the area of the grey overlay represents the ratio of stored hyporheic water to discharged hyporheic water over a period of 1 h. Note that the depicted time-scale of discharge, in this case 1 h, is discernible from the plot; by definition, the grey overlay crosses the circle perimeter at the depicted time-scale.

**Table 1. Some reported values of $\alpha$ based on experimental tracer releases.**

| Source | $\alpha$ | Stream Name | Description |
|---|---|---|---|
| Haggerty et al., 2002 [25] | 1.28 | (not reported)[1] | 2nd-order mountain stream |
| Gooseff et al., 2005 [40] | 1.28 | WS03[1] | 2nd-order stream reach with extensive colluvial and entrained alluvial fill and step-pool morphology |
| Gooseff et al., 2003 [41] | 1.30 | WS03[1] | 2nd-order, single-thread, tightly spaced pool-step morphology, 12.6% avg. gradient |
| | 1.53 | LO410[1] | 4th-order, single-thread alluvial channel with widely-spaced step-pool/step-riffle morphology, 4.84% avg. gradient |
| | 1.58 | LO411[1] | 4th-order, braided alluvial channel that terminates at a channel-spanning bedrock outcrop, 4% avg. gradient |
| Gooseff et al., 2007 [42] | 1.87 | Headquarters Stream[2] | agricultural stream flowing through irrigated grazing land with grassy banks |
| | 1.74 | Ditch Creek[2] | natural |
| | 1.89 | Two Ocean Creek[2] | natural |
| Drummond et al., 2012 [43] | 1.35 | Säva Stream[3] | markedly vegetated with emerged and submerged macrophytes, sediment consists mainly of clay, but at the upper part sediment consists of silt, gravel, and detritus particles of differing sizes |

[1]H.J. Andrews Experimental Forest, Oregon, USA

[2]Jackson Hole, Wyoming, USA

[3]Uppsala, Uppsala County, Sweden

## Illustrative applications

**A visualization from conservative tracer data.** To create a visualization of patterns of exchange in a simple, real-world hyporheic system, we constructed an annular flume by nesting a cylindrical 38 cm diameter food-grade polyethylene tank within a similar 56 cm diameter tank, thus creating a 9 cm wide circular "raceway" approximately 40 cm deep (Fig 6a). We added glass beads (1.2–1.6 mm in diameter) to a depth of 15 cm within the raceway and then graded the beads around the raceway to form a single sinusoidal dune, 20 cm thick on one side of the flume and 10 cm deep on the opposite side. We then filled the raceway with 25 l of water, yielding to a combined water- and glass-bead depth of 30 cm. Flow around the raceway (velocity of $\sim 10$ cm s$^{-1}$) was induced using by the jet from a submersible impeller-driven aquarium pump.

We removed 100 ml of water from the flume, added 1g NaCl to the sample, and reintroduced the salt solution to the flume as a slug. We monitored specific conductance in the "channel" (surface) water ($K_c$) of the flume until $K_c$ came into equilibrium with the the specific conductance of "hyporheic" (interstitial) water ($K_h$) within the glass beads. The rate a which $K_c$ reached equilibrium was mediated by hyporheic exchange induced by water flow over the dune. The observed $K_c$ prior to the addition of the salt slug ($K_x$) was 248.4 $\mu$S cm$^{-1}$, the $K_c$ immediately following the salt slug addition ($K_0$) was 383.6 $\mu$S cm$^{-1}$, and the $K_c$ at equilibrium ($K_\infty$) was 356.9 $\mu$S cm$^{-1}$.

We assumed that specific conductance was proportional to salt concentration; therefore, we treated values of specific conductance as the concentration of a non-reactive solute. Considering conservation of mass, the average value of $K_c$ and $K_h$, weighted by $V_c$ and $V_h$ respectively, is at all times after the slug injection equal to $K_\infty$. At the time of slug addition, this equality can be represented as:

$$K_\infty = \frac{V_c}{V}(K_0 - K_x) + K_x, \tag{13}$$

where V is the total volume of water added to the flume (25 l). We calculated the volume of

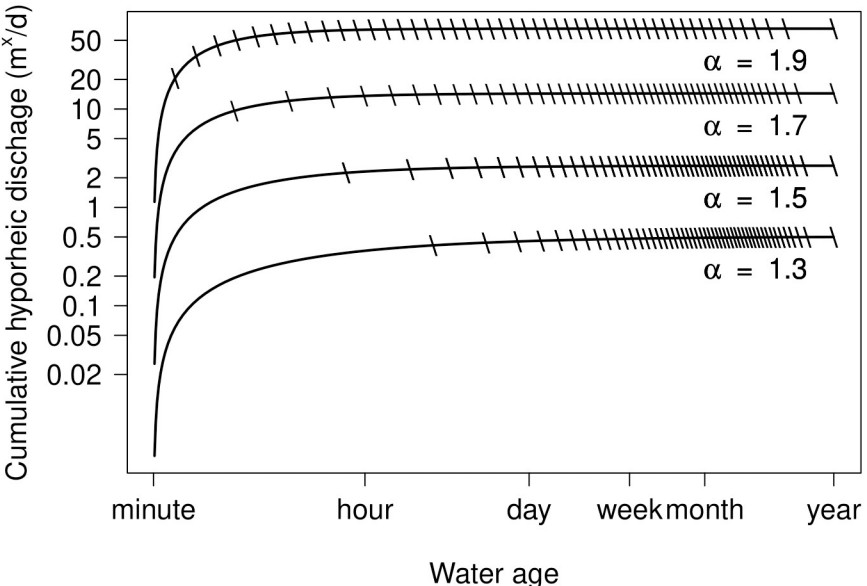

**Fig 4. Cumulative hyporheic discharge to the channel (log scale) by water age (log scale) for a representative unit (RU) of water stored in a hyporheic zone.** Each curve is associated with a different value of $\alpha$, the negative power-law exponent used to describe the hyporheic water age distribution of hyporheic discharge. Units of hyporheic exchange (y-axis) can be interpreted in terms of length (e.g., m day$^{-1}$), area (e.g., m$^2$ day$^{-1}$), or volume (e.g., m$^3$ day$^{-1}$) depending on whether the RU is one dimensional (e.g., 1 m of hyporheic water thickness), two dimensional (e.g., 1 m$^2$ of hyporheic water cross-sectional area), or three dimensional (e.g., 1 m$^3$ of hyporheic water). Hash marks on each curve demarcate the maximum water age ($\tau_i$) for each of 50 transient storage zones (TSZs) in the RU; each TSZ contains 2% (0.02 m$^x$) of water stored in the RU.

surface water in the flume ($V_c$) empirically, by rearranging Eq 13:

$$V_c = V * (K_\infty - K_x)/(K_0 - K_x). \tag{14}$$

The volume of hyporheic water ($V_h$) is simply $V - V_c$. Using Eq 14, $V_c$ = 20.06 l and $V_h$ = 4.94 l.

Given estimates of $V_c$ and $V_h$ and again considering conservation of mass, the value of $K_c$ for any elapsed time since the slug release ($t$) can be determined from the mean $K_h$ of hyporheic water:

$$K_{c_t} = \frac{VK_\infty - V_h K_{h_t}}{V_c}, \tag{15}$$

Hyporheic geometry—specifically the internal age density function ($I(\tau)$)—can be used to estimate the mean value of $K_{h_t}$ as surface water mixes with hyporheic water by assuming that hyporheic water maintains the same conservative tracer concentration it had when it entered the hyporheic zone:

$$K_{h_t} = \int_{\tau=\tau_0}^{\tau_n} I(\tau)\, K_{c_{t-\tau}}\, d\tau \tag{16}$$

Note that Eq 16 neglects the effects of dispersion within the hyporheic zone, a simplification that becomes increasingly problematic in natural streams, where hyporheic systems have greater water ages.

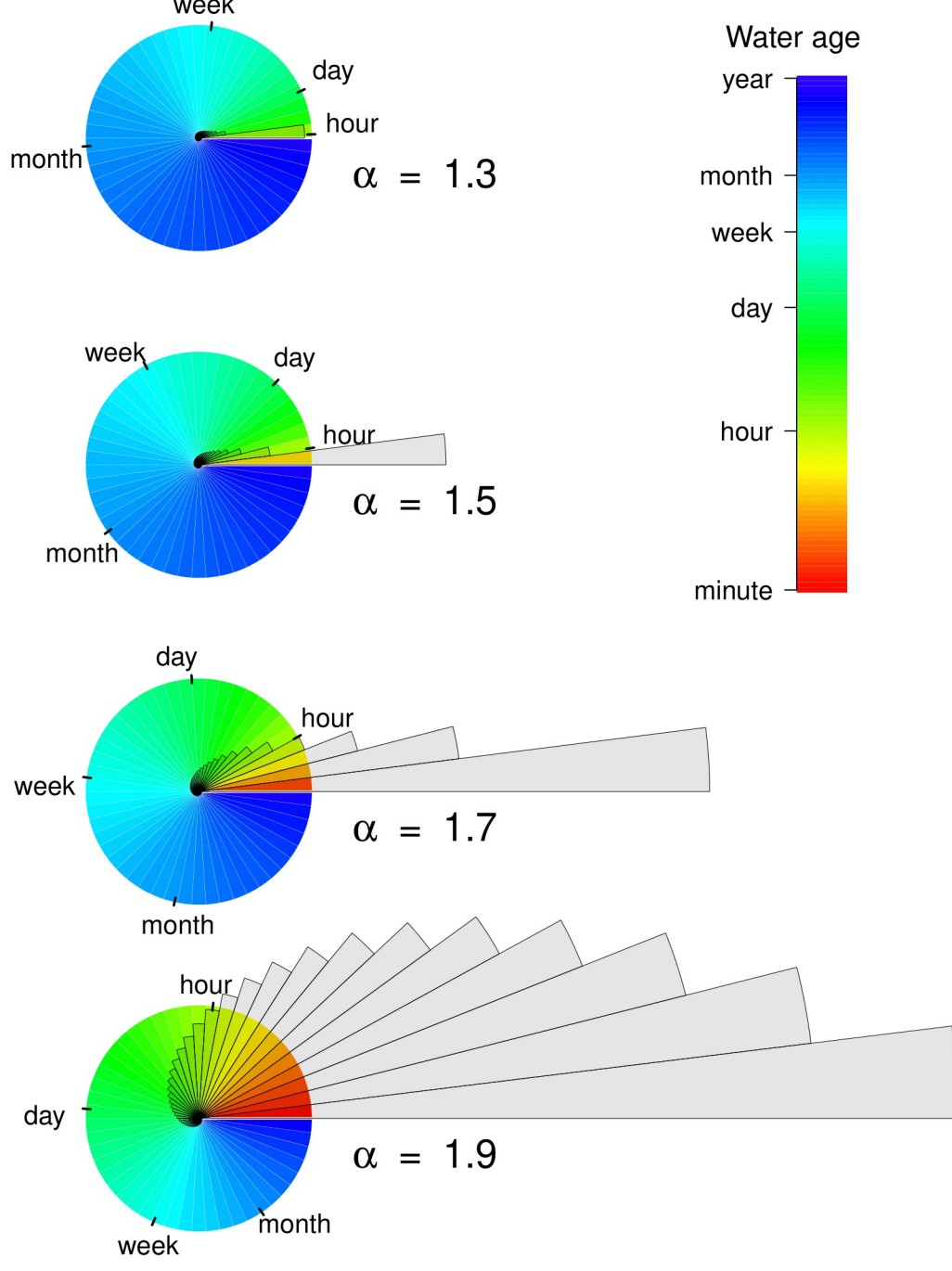

**Fig 5. Quantitative depiction of hyporheic hydraulic geometry for different values of $\alpha$.** Pie chart represents 50 transient storage zones (TSZs) within a representative unit (RU; 1 m$^x$) of hyporheic water stored within the hyporheic zone. Each TSZ contains 2% (0.02 m$^x$) of the RU's water storage; color represents the mean water age of each TSZ. The area of superimposed grey wedges is proportional to the water units discharged to the channel from each TSZ in a 1 h period. The 1 h time-scale of depicted discharge can be inferred from plot because, at $\tau = 1$ h, the storage and discharge wedges are equal in area. The nautilus shaped distribution of grey wedges describes the relative water discharge from each TSZ, while the color distribution represents water age across TSZs within the hyporheic zone.

We wrote code in the R statistical computing environment [44] to solve Eq 15 and a finite difference approximation of Eq 16 via iteration over time, with a time step representing 1/1000 of the duration between $\tau_0$ and $\tau_n$. We assumed a value of 1 s for $\tau_0$ and used the code to fit values of $\tau_n$ and $\alpha$ (nested within $I(\tau)$ in Eq 16) to the observed values of $K_c$ from the annular flume (Fig 6b). The agreement between modeled and observed data suggested a power law was a useful approximation of the shape of the $AD_d$ of the dune in the flume. Resulting parameter estimates were $\tau_n$ = 4337 s and $\alpha$ = 1.70. Based on $\tau_0$ = 1 s and fitted values of $\tau_n$ and $\alpha$, the relationship between storage, water age, and hyporheic discharge in the flume is shown in Fig 6c.

**Scaling hyporheic effects on water temperature.** To further illustrate the utility of considering hyporheic geometry, we use $E(\tau)$ and $I(\tau)$ to consider how variation in $\alpha$ might influence temperature dynamics in water stored within the hyporheic zone vs. in water discharged from the hyporheic zone. For this simple application, we offer the concept of a "representative hyporheic flow path"—conceptually, a flow path that reflects how water temperature typically varies with water age in the aquifer (Fig 7). As water traverses the representative flow path and water age increases, daily and seasonal variation typical of the channel are lagged and damped relative to the stream channel [45, 46]. At the beginning of the flow path, high-frequency diel temperature patterns damp quickly. Farther along the flow path, low-frequency annual temperature signals are damped.

We calculated the mean temperature of water stored within the hyporheic zone ($\bar{T}_s$) as:

$$\bar{T}_{s,t^*} = \int_{\tau_0}^{\tau_n} T(t^*, \tau) I(\tau) \, d\tau. \tag{17}$$

and the mean temperature of hyporheic discharge as:

$$\bar{T}_{\uparrow,t^*} = \int_{\tau_0}^{\tau_n} T(t^*, \tau) E(\tau) \, d\tau, \tag{18}$$

where $T(t, \tau)$ is a function that returns hyporheic water temperature at a specified time of the year ($t^*$) for a specified water age ($\tau^*$)—specifically the hyporheic temperature dynamics represented in Fig 7. Because both $I(\tau)$ and $E(\tau)$ are PDFs, Eqs 17 and 18 simply calculate a weighted average of the temperatures along the flow path, where the PDFs provide the weights. $I(\tau)$ represents the $AD_s$—the relative amount of hyporheic water storage across water ages. $E(\tau)$ represents the $AD_d$—the relative amount of hyporheic discharge across water ages. Fig 8 shows the results of a simple model representing Eqs 17 and 18, plotted atop the channel temperature pattern used to drive the model.

## Discussion

Fig 4 reveals several important aspects of the expected change in hyporheic hydrology associated with variation in $\alpha$. Remembering that our conceptual model assumes steady state flow—i.e., recharge of the hyporheic zone from the channel ($q_\downarrow$) is equal in magnitude to the sum of hyporheic discharge from all TSZs ($q_\uparrow$)—the terminus of each cumulative distribution in Fig 4 represents the rate of hyporheic exchange (magnitude of $q_\downarrow$ and $q_\uparrow$) associated with each value of $\alpha$. Thus, Fig 4 shows how each increase in $\alpha$ yields a substantial increase in the rate of hyporheic exchange through the RU; as values of alpha increase from 1.3 to 1.9, total hyporheic exchange per unit hyporheic water storage spans more than 3 orders of magnitude. Additionally, Fig 4 shows that $\alpha$ influences the fraction of the RU from which water of different ages emerges. For instance, remembering that all TSZs were defined to contain 0.02 m$^x$ (2%) of the water storage in the RU, Fig 4 shows that the first TSZ ($TSZ_1$) discharges water with an age

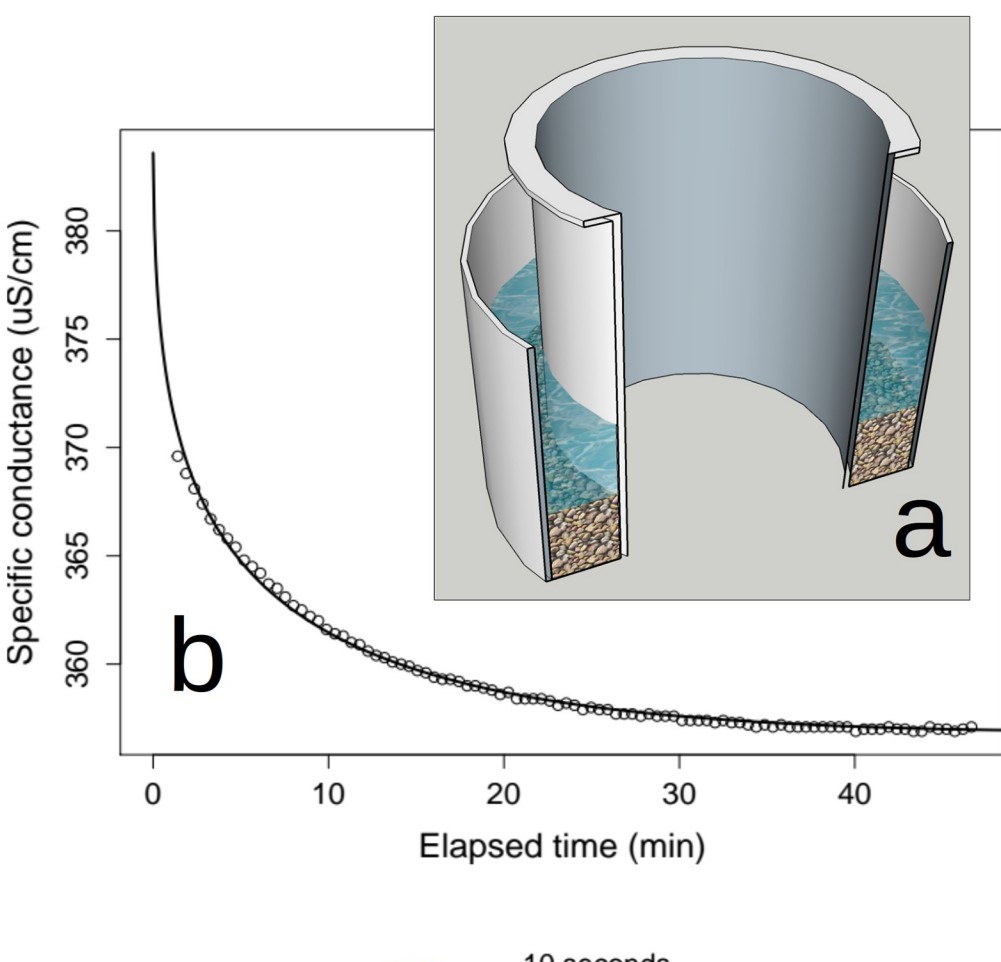

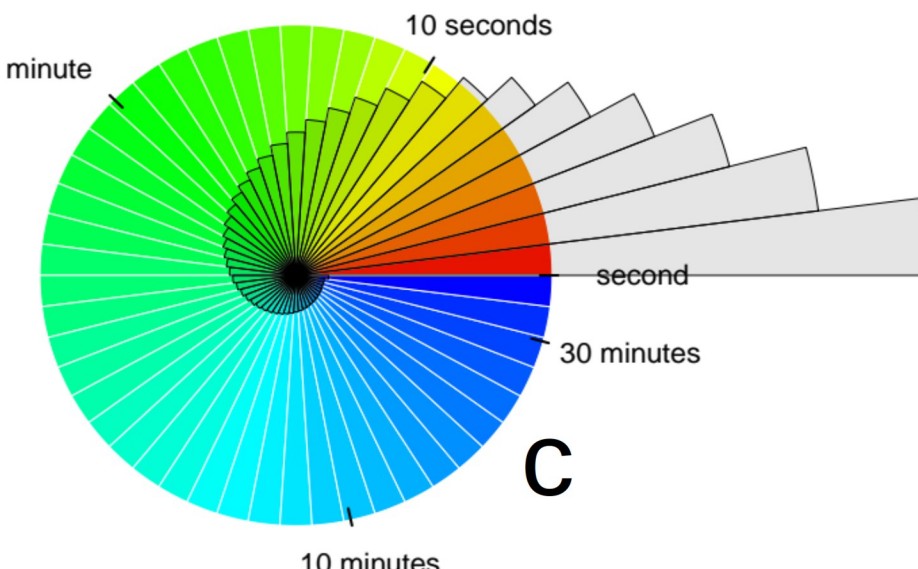

**Fig 6. Visualization of hyporheic water exchange in an annular flume.** (a) Cut-away diagram of the annular flume. (b) Observed (points) and modeled (line) surface water specific conductance. Modeled data derived by fitting values of $\tau_n$ and $\alpha$ to observed data using Eqs 15 and 16. (c) Visualized relationship among water age, hyporheic exchange, and interstitial water storage in the flume. Grey wedges represent aquifer discharge rates for a period of 10 seconds.

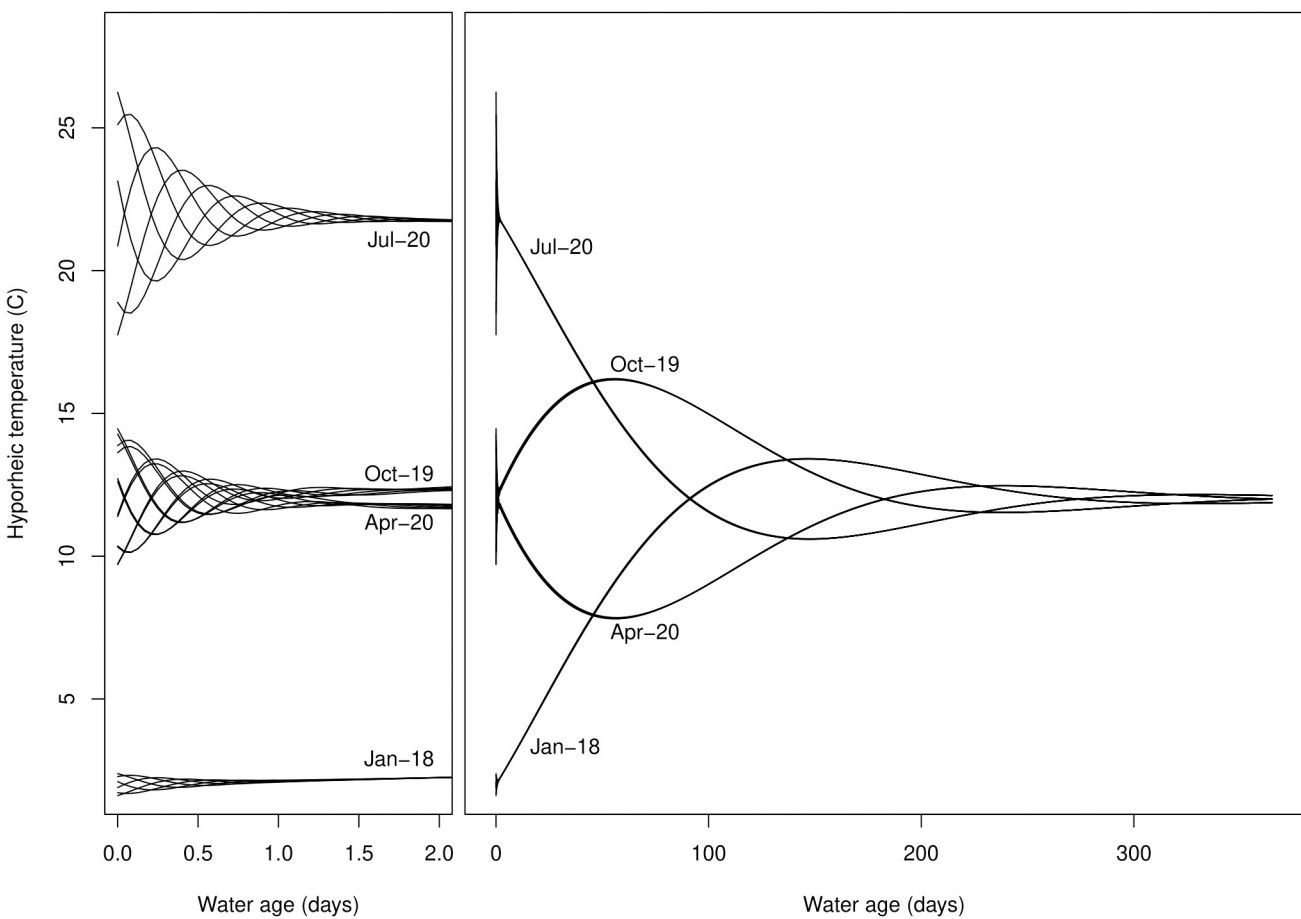

**Fig 7. Characteristic temperature damping and lagging with water age in an expansive hyporehic zone based on relationships presented by Helton et al. [47].** Lines represent idealized patterns of temperature along a "representative flow path" through an expansive coarse-grained alluvial aquifer for four different dates (approximate annual maximum, minimum, and mean stream temperatures) as a function of water age. Each line on the daily temperature plot (left) represents the expected pattern of temperature variation for a different hour of the day. Daily temperature variation damps quickly with water age. Seasonal variation in temperature (right) is visible at greater water ages.

between 60 s to 2.4 h when $\alpha$ = 1.3. When $\alpha$ = 1.9, water with an age between 60 s to 2.4 h flows through roughly 15 TSZs ($\sim$ 30% of the aquifer).

Our graphical depictions in Fig 5 illustrate this same concept in a different way. The rate of hyporheic exchange per unit volume of hyporheic water storage (proportional to the size of grey wedges in Fig 5) increases markedly as $\alpha$ increases from 1.3 to 1.9. Logically, this pattern must be true. As $\alpha$ increases from 1.3 to 1.9, the water age of hyporheic discharge, $AD_d$, skews toward younger water, and thus a hyporheic zone with a younger mean water age. In order for the mean of the $AD_d$ for an RU to skew younger, flow through the fixed volume must increase. Importantly, however, regardless of the value of $\alpha$ and the associated rate of hyporheic exchange, the bulk of hyporheic exchange is associated with younger water age.

Fig 5 also illustrates that, regardless of the value of $\alpha$, the $AD_d$ ($\tau$ of water discharged to the channel) is more heavily skewed toward young water ages than the $AD_s$ ($\tau$ of stored hyporheic water). In other words, there is a greater proportion of older hyporheic water stored than the proportion that is discharged. Such a difference in the distribution of $\tau$ for the $AD_d$ vs. the $AD_s$ has several important implications for ecosystem dynamics in streams. Specifically, the $AD_d$ is

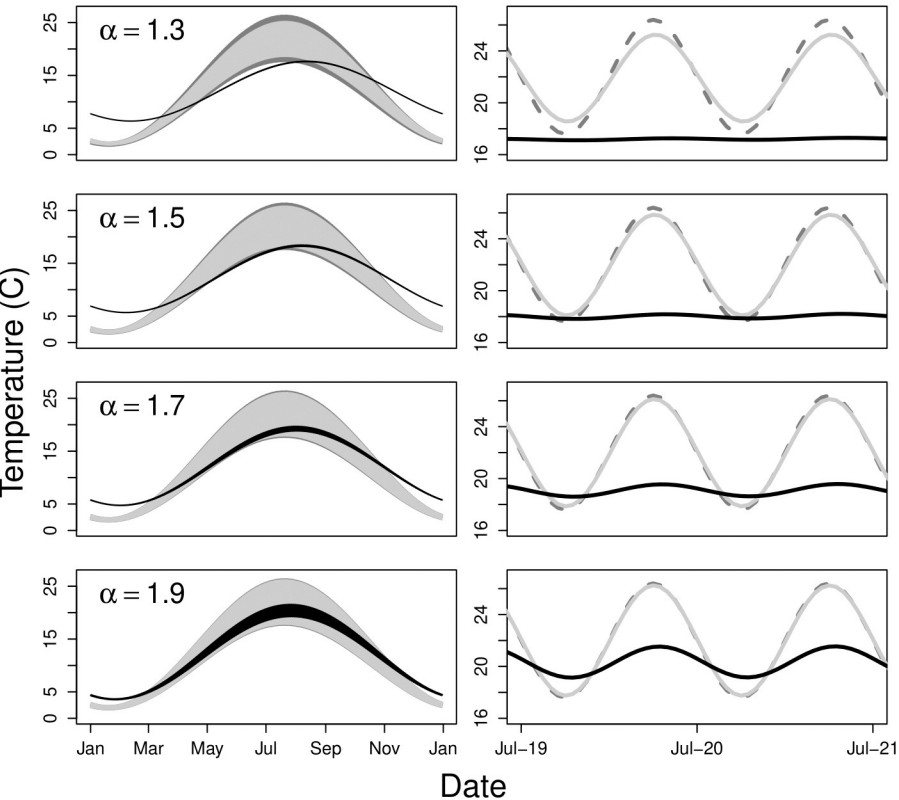

**Fig 8. Simulated patterns of mean water temperature within the channel, the alluvial aquifer and upwelling from the aquifer.** Simulated patterns of mean temperature for water discharged from the alluvial aquifer (light grey) and stored within an idealized alluvial aquifer (black) plotted with channel water temperature (dark grey; dashed) over time for different values of $\alpha$.

important for understanding how hyporheic discharge influences channel water characteristics. In contrast, the $AD_s$ of hyporheic water storage governs the relative contribution of hyporheic processes (e.g., productivity, metabolism, nutrient cycling) to the whole stream ecosystem. Therefore, the differences between the $AD_d$ and the $AD_s$ suggest that processes occurring in young hyporheic water are likely to drive the influence of hyporheic exchange on surface water. In contrast, processes associated with advanced water age will be important drivers of ecological dynamics occurring within hyporheic zones themselves.

Importantly, the discussion to this point has considered a hyporheic zone with fixed storage—a representative unit of the hyporheic zone. Yet, water storage in a hyporheic zone is not static, but changes with variation in surface discharge [5, 48]. Water storage in hyporheic zones is a function of channel stage, sediment properties (e.g, hydraulic conductivity and porosity) and alluvial aquifer size. Eq 10 highlights the importance of the size of the hyporheic zone in determining the magnitude of $q_\downarrow$. Specifically, for any given value of $\alpha$, $q_\downarrow$ is directly proportional to hyporheic storage volume.

Our work suggests that research efforts to identify stream characteristics correlated with values of $\alpha$ and $s$ would expand opportunities for rapid characterization of hyporheic hydrology across stream networks [49, 50], especially if such correlates were obtainable from readily available spatial data sets (digital elevation models, LIDAR, aerial photography, etc.). Prior research [1, 18, 37] has shown that streams with self-organized geomorphic patterns at

multiple scales (e.g., streambed dunes, well organized pool-riffle sequences, side channels, greater sinuosity—hereafter "geomorphic complexity") are associated with higher rates of hyporheic exchange, suggesting that geomorphic analysis may be an important starting point in attempts to estimate $\alpha$ and $s$. For instance, if more complex channels have higher rates of hyporheic exchange, such channels are likely to be associated with larger values of $\alpha$ (assuming $s$ is similar, which is a generous assumption). Similarly, processes or management actions that reduce sinuosity, side channels, and the complexity of bed-forms alter hyporheic water age distributions. Such actions generally reduce hyporheic exchange rates and increase hyporheic water ages, yielding smaller values of $\alpha$ with associated implications for water temperature, nutrient dynamics, and other aspects of hyporheic ecology.

Complicating this picture, of course, is the fact that changes in river stage are likely to yield values of $\alpha$ and $s$ that are dynamic over time within a single stream reach [48]. Thus, the internal distribution of water age is apt to vary with river stage [30], perhaps especially in systems with seasonally inundated side-channels [5]. Although we discuss $\alpha$ and $s$ somewhat independently, the two values may be interdependent. Large hyporheic zones (with higher $s$) are likely to allow greater opportunity for development of long flow paths, which may be associated with an increase in the frequency of long flow paths and an associated reduction in $\alpha$. Although logical when considering the equations presented here, our suggestion of a relationship between $\alpha$ and $s$ is speculative. Yet if such relationships exist, the challenge of estimating $s$ and $\alpha$ across basins may be somewhat more tractable than if the two values are relatively independent of one another.

The results from our simple flume experiment (Fig 6) yielded detailed information about the nature of hyporheic exchange within the glass-bead dune in the flume. Using Eq 10 and fitted values of $\tau_n$ and $\alpha$, we can conclude that the rate of hyporheic exchange within the flume for water age $1 \text{ s} < \tau < 4337 \text{ s}$ was approximately $0.23 \text{ l s}^{-1}$. Dividing the 4.93 l of hyporheic water storage by the exchange rate yields $\sim 21$ s as the mean water age of hyporheic discharge. Yet the empirically derived estimate of maximum water age in the system ($\tau_n = 4337$ s) indicated that some of the flume's hyporheic water required more than an hour to exchange with surface water. Such discrepancies between mean and maximum water age underscore the tailed shape of the flume's exit function ($E(\tau)$). In fact, applying $E(\tau)$ with fit values of $\tau_n$ and $\alpha$ reveal that only 10% of water entering the hyporheic zone remained in the flume's hyporheic zone for longer than than 23 s, 40% exited the hyporheic with a water age between 2.6s and 23 s, and fully 50% of hyporheic exchange exited with a water age between 1 and 2.6 s. These water age values may seem surprisingly skewed, yet the fitted value of $\alpha$ (1.70) for the flume is well within the range of observed values from field experiments in natural streams (Table 1).

Our analysis of stream temperature illustrates the importance and benefits of considering hyporheic hydraulic geometry. Field studies reveal that hyporheic zones contain habitats with exceptionally diverse thermal, biogeochemical, and biological conditions [47, 51, 52]. Considering the age distributions of water discharge and storage within the hyporheic zone provides the potential for a quantitatively rigorous but practical mechanism for scaling hyporheic heterogeneity to whole stream networks. Importantly, however, the results in Fig 8 are intended to be illustrative rather than predictive. Our application of Eqs 17 and 18 is oversimplified. First, $T(t^*, \tau)$ is held constant across values of $\alpha$ when, in fact, changing the rate of hyporheic exchange will alter the patterns of temperature damping and lagging in the aquifer [45]. Further, our application shows patterns of hyporheic temperatures for a fixed stream water temperature regime (plotted in Fig 8); we do not consider the feedback of returning hyporheic water altering stream channel temperature, which in turn would affect hyporheic temperature. Regardless, Eqs 17 and 18 illustrate conceptually how considering the hydraulic geometry of the hyporheic zone (in this case, $E(\tau)$ and $I(\tau)$) can be used to scale hyporheic water

characteristics from flow-paths to floodplains, given a value for $\alpha$ and any linear or non-linear empirical or mechanistic relationship between $\tau$ and a water characteristic of interest (e.g., temperature (Fig 7), nutrient concentrations, microbial community composition) in the hyporheic zone.

Finally, our assumption of a power-law $AD_d$ [25, 31, 37, 53] underscores a subtle and important fact: the rate of hyporheic exchange reported for any given stream is dependent on the range of water ages considered. In our equations, values of $\tau_0$ and $\tau_n$ create distinct although admittedly artificial demarcations between channel-, hyporheic-, and ground-water where no such clear distinction exists. The choice of $\tau_0 > 0$, for instance, suggests that water spending less time than $\tau_0$ in the streambed is considered to have never left the channel. While use of $\tau_0$ in this manner might seem unappealing (interpreted as yielding an underestimate of the "true" hyporheic exchange), the opposite assumption ($\tau_0 \approx 0$) is no less cumbersome. As $\tau_0$ becomes vanishingly small, the volume of water that merely contacts the streambed surface for a fraction of a second would constitute a very high if not inflated rate of "hyporheic exchange". We therefore suggest that—regardless of the assumed $AD_d$ shape—the minimum duration of interaction with the streambed that constitutes "hyporheic water" may be a non-trivial consideration.

Consider, for instance, the fact that power-law scaling may break down for very low water ages [54]. Such a result might be expected, given the fractal nature of nested hyporheic flow paths [55]. Just as the measured length of any coastline is dependent upon the scale of measurement, the magnitude of any empirical or modeled estimate of $q_{\downarrow}$ must be dependent upon some inherent minimum time-scale of water age. It follows, then that reported estimates of hyporheic exchange rates in the literature are dependent upon the time-scale of water ages considered or measured, although this is seldom acknowledged and the timescale that applies to a given estimate of hyporheic exchange typically is neither pondered nor reported. For instance, three-dimensional finite element or finite difference floodplain models can simulate hyporheic exchange as driven by stream discharge regime and stream morphology. The choice of cell size or node spacing within such a model sets the lower limit on the length of the flow path (and therefore the lower limit of water age) that can be considered within the model. Thus, any estimate of hyporheic exchange rate from such a model is bounded by the water age associated with the finest spatial scale at which geomorphic features can be represented within the model. Similarly, a seepage meter [56] placed to estimate rates of exchange can not account for exchange that would otherwise occur within the bed area encompassed by the meter. As another example, rates of hyporheic exchange from tracer experiments are typically estimated from the tail of the breakthrough curve; short time-scale hyporheic exchange is largely indistinguishable from in-channel transient storage that occurs on the same time scale.

Thus, we offer caution against the assumption that there is a single, time-scale independent rate of hyporheic exchange within any stream channel. We argue that any reported rate of hyporheic exchange is associated with an implicit range of water age time-scales, likely to range from $> 0$ to $< \infty$. In essence, given the tailed distribution of water-age typical of hyporheic zones, any two methods of estimating $q_{\downarrow}$ that consider two different time-scales of water age will yield two different estimates of $q_{\downarrow}$, even if applied contemporaneously to the same stream.

While the time-scale dependence of hyporheic exchange may seem somewhat intractable, acknowledging this aspect of hyporheic exchange can be rather straightforward. In empirical studies, we encourage researchers to think carefully about the minimum and maximum length- or time-scales captured by their experiments, and report them. In modeling experiments, the choice of $\tau_0$ (for 1-D models) or the density of landscape tessellation (for 3-D models) can be carefully considered in relation to the model's purpose. As an example, we can

build on the water temperature application we presented earlier. Specifically, the magnitude of water temperature change at a hyporheic water age of 60 s will seldom be measurable. Therefore, $\tau_0 = 60$ s is an appropriate value for applications concerned with understanding hyporheic water temperature. We believe that future experiments designed to characterize hyporheic hydrology and associated physical, chemical, and biological properties will benefit from more mindful and rigorous consideration of the inter-dependency between hyporheic exchange magnitude and the range of water age timescales inherent in the design of simulation or empirical studies of hyporheic hydrology.

## Conclusion

Assuming we accept the assumption that the $AD_d$ of hyporheic zones is asymmetrical and tailed, the mathematical linkage between $s$, $\tau$, and $q_\downarrow$ in hyporheic zones is defined by the shape of the $AD_d$. In the case of a power-law representation, this shape is determined by the value of $\alpha$, the exponent of the power law (Eq 11). As $\alpha$ increases, the shape of a power law plot (e.g., Fig 3) becomes more concave and the $AD_d$ of water exiting the hyporheic zone becomes more skewed toward younger water ages. If the volume of the hyporheic zone is held constant, a skew toward younger water ages requires an increase in the rate of flow through the hyporheic zone (e.g., the rate of hyporheic exchange). Conceptually, then, the inherent inter-relationships among $s$, $\tau$, and $q_\downarrow$ per unit volume of hyporheic zone yield the patterns shown in Figs 4 and 5. Specifically, when $\alpha$ increases, the rate of hyporheic exchange increases, shifting the $AD_d$ and $AD_s$ toward younger water.

The equations in this paper, from which our conclusions are drawn, are presented in a manner intended to be as transparent as possible. While grasping the math improves understanding, the visualizations alone are sufficient to illustrate the following key points:

- the $AD_d$ is not the same as the $AD_s$ because the $AD_d$ represents $\tau$ at the ends of hyporheic flow paths while $AD_s$ represents $\tau$ along the length of hyporheic flow paths;

- the $AD_d$ and $AD_s$ are linked mathematically—each can be derived from the other;

- the $AD_d$ (and to a lesser extent, the $AD_s$) is heavily skewed toward young water ages and thus the majority of hyporheic exchange traverses brief flow paths;

- the $AD_d$ is important for predicting hyporheic influences on surface water while the $AD_s$ is useful for characterizing and scaling heterogeneity within the hyporheic zone itself;

- with our simplified approach and an assumption of a power-law, one variable ($\alpha$) is sufficient to characterize the $AD_d$ and $AD_s$ while addition of a second variable ($s$) allows estimates of hyporheic exchange rates; and

- $\tau_0$ and $\tau_n$ are not arbitrary factors that can cause over- or under-estimates some "true" value of $q_\downarrow$. Rather, they represent the bounds of the water age timescales to which any empirical or modeled estimate of $q_\downarrow$ applies.

By employing the established mathematical notation of chemical engineers, we intend to promote a view of streams and their associated hyporheic zones as "natural bioreactors." Adopting such a view provides a simple mechanism for scaling field observations of hyporheic metabolism, biogeochemistry, and temperature to whole stream ecosystems. Finally, we hope our characterization of hyporheic hydraulic geometry will help engender broader appreciation for the shape of hyporheic water age distributions and associated implications for the ecology of running waters.

## Supporting information

**S1 Appendix.**
(TEX)

## Acknowledgments

We are grateful to four anonymous reviewers for extensive comments on drafts of this manuscript.

## Author Contributions

**Conceptualization:** Geoffrey C. Poole, S. Kathleen Fogg, Scott J. O'Daniel, Byron E. Amerson, Ann Marie Reinhold, Samuel P. Carlson.

**Formal analysis:** Geoffrey C. Poole.

**Funding acquisition:** Geoffrey C. Poole, Scott J. O'Daniel.

**Investigation:** Geoffrey C. Poole, S. Kathleen Fogg.

**Methodology:** Geoffrey C. Poole, S. Kathleen Fogg, Ann Marie Reinhold, Samuel P. Carlson.

**Project administration:** Geoffrey C. Poole, Scott J. O'Daniel.

**Resources:** Geoffrey C. Poole.

**Supervision:** Geoffrey C. Poole.

**Validation:** Geoffrey C. Poole, S. Kathleen Fogg.

**Visualization:** Geoffrey C. Poole, S. Kathleen Fogg, Byron E. Amerson, Ann Marie Reinhold, Samuel P. Carlson, Elizabeth J. Mohr, Hayley C. Oakland.

**Writing – original draft:** Geoffrey C. Poole, S. Kathleen Fogg.

**Writing – review & editing:** Geoffrey C. Poole, S. Kathleen Fogg, Scott J. O'Daniel, Byron E. Amerson, Ann Marie Reinhold, Samuel P. Carlson, Elizabeth J. Mohr, Hayley C. Oakland.

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
