## [Decision Letter · Decision Letter 0]

24 Jun 2021

PONE-D-21-04355

Hyporheic hydraulic geometry

PLOS ONE

Dear Dr. Fogg,

Thank you for submitting your manuscript to PLOS ONE. After careful consideration, we feel that it has merit but does not fully meet PLOS ONE’s publication criteria as it currently stands. Therefore, we invite you to submit a revised version of the manuscript that addresses the points raised during the review process. More specifically, I encourage you to focus on the three main concerns of Reviewer 3 on novelty, application of examples and clarification of concepts, and the four main concerns from Reviewer 2.

We look forward to receiving your revised manuscript.

Kind regards,

Clara Mendoza-Lera

Academic Editor

PLOS ONE

Journal Requirements:

2. Please modify the title to ensure that it is meeting PLOS’ guidelines (https://journals.plos.org/plosone/s/submission-guidelines#loc-title). In particular, the title should be "specific, descriptive, concise, and comprehensible to readers outside the field" and in this case it is not informative and specific about your study's scope and methodology

4. We note that Figure 2 in your submission contain map/satellite images which may be copyrighted. All PLOS content is published under the Creative Commons Attribution License (CC BY 4.0), which means that the manuscript, images, and Supporting Information files will be freely available online, and any third party is permitted to access, download, copy, distribute, and use these materials in any way, even commercially, with proper attribution. For these reasons, we cannot publish previously copyrighted maps or satellite images created using proprietary data, such as Google software (Google Maps, Street View, and Earth). For more information, see our copyright guidelines: http://journals.plos.org/plosone/s/licenses-and-copyright.

4.1.    You may seek permission from the original copyright holder of Figure 2 to publish the content specifically under the CC BY 4.0 license. 

4.2.    If you are unable to obtain permission from the original copyright holder to publish these figures under the CC BY 4.0 license or if the copyright holder’s requirements are incompatible with the CC BY 4.0 license, please either i) remove the figure or ii) supply a replacement figure that complies with the CC BY 4.0 license. Please check copyright information on all replacement figures and update the figure caption with source information. If applicable, please specify in the figure caption text when a figure is similar but not identical to the original image and is therefore for illustrative purposes only.

Reviewers' comments:

Reviewer's Responses to Questions

**Comments to the Author**

1. Is the manuscript technically sound, and do the data support the conclusions?

Reviewer #1: Partly

Reviewer #2: Partly

Reviewer #3: Partly

Reviewer #4: Yes

2. Has the statistical analysis been performed appropriately and rigorously? 

Reviewer #1: N/A

Reviewer #2: N/A

Reviewer #3: N/A

Reviewer #4: Yes

3. Have the authors made all data underlying the findings in their manuscript fully available?

Reviewer #1: No

Reviewer #2: Yes

Reviewer #3: Yes

Reviewer #4: Yes

4. Is the manuscript presented in an intelligible fashion and written in standard English?

Reviewer #1: Yes

Reviewer #2: Yes

Reviewer #3: Yes

Reviewer #4: Yes

5. Review Comments to the Author

Reviewer #1: This manuscript analyzes travel-time distributions in the hyporheic zone under the assumptions of steady state and negligible net gain or loss of groundwater. The authors emphasize that the age distribution of the water returning to the stream differs from the age distribution within the hyporheic zone. However, this has already been known. I had really hard times figuring out what's new in the manuscript.

The authors motivate their study by the analysis of expansive coarse-grained alluvial aquifers (for which they introduce the completely unnecessary acronym ECGAA). I doubt that the hydrology of these large gravel bodies is ever at steady state, and I also doubt that neglecting any net groundwater upwelling or permanant loss of streamwater to groundwater is appropriate. However, under the given restrictions, the relationship between travel times (also denoted transit times) and groundwater age can be looked up in Liao & Cirpka (2011, doi: 10.1029/2010WR009927, Appendix A), which is most likely not the first such derivation. Note that a formulation of hyporheic exchange that are based on transit/travel-times have already been proposed by Wörman et al. (2002, doi:10.1029/2001WR000769) if not much earlier, whereas formulations using the age distribution/memory function are at the heart of multi- (or single-)rate mass transfer formulations. Under the given conditions, you get one from the other by taking derivatives or integration. To complete the description of exchange, you either need the volume of the hyporheic zone (relative to the volume of the stream), or the exchange-rate coefficient. Period.

The notation preferred by the authors is confusing. The term residence time distribution is used by other authors as the age distribution. Please read the landmark paper of Botter et al. (2011, doi:10.1029/2011GL047666). This paper also discusses the impacts of transient flow (for catchments rather than the hyporheic zone, but their master equation could of course be adapted). It would be interesting to see how the concept of StorAgeSelection functions (Rinaldo et al., 2015, doi: 10.1002/2015WR017273) could be used to describe systems with extensive hyporheic exchange. But obviously, the authors are not aware of these concepts.

Another odd term used by the authors are "isotemporal surfaces". These guys are known as isochrones, a very established concept in hydrology.

As application, the authors chose the temperature regime of the extended gravel plains. Too bad, heat is the one extensive state variable in shallow groundwater for which I would definitely NOT rely on advective travel times. Conduction into the transverse direction leads to a comparably strong thermal exchange among streamtubes, causing solute travel times to be different from "temperature travel times", and even worse making heat exchange with the land surface a relevant process. The conceptual model of the authors is that the hyporheic zone is thermally isolated from the land surface, that transverse exchange can be neglected, and that the classical Stallman (JGR 1965) solution on the one-dimensional propagation of sinusoidal temperature variations holds. You'd better start with the 3-D heat balance equation with realistic boundary conditions. It is very likely that the diurnal signal of temperature cannot be explained by the travel times of solutes plus the Stallman solution.

Let me finish with my bewilderment on the statement "that integrals and probability density functions are typically presented more thoroughly in engineering curricula than in ecological curricula" (lines 42-43). What type of an argument is that? You can expect basic calculus and probability theory from any quantitative scientist (and I would add linear algebra, vector calculus, and differential equations to the mix). Otherwise it's not science. However, if the authors want to address readers without any math background, their equations won't help either.

Reviewer #2: In this study, Poole et al. use chemical engineering reactor theory to quantify residence time distribution (RTD) and the distribution of water ages for a conceptual, coarse sediment hyporheic zone. Their specific aims for this work are to introduce the theory to fluvial ecologists and resource managers, as well as highlight the relation between various metrics that describe the hyporheic zone and surface water-groundwater interactions. For the latter aim, the authors present various figures that demonstrate the relation between RTD, water age in the hyporheic zone, hyporheic exchange fluxes, and hyporheic zone storage volume for a simple alluvial aquifer parameterized with a power-law RTD. They discuss the assumptions underlying their modeling exercise.

There is no new mathematical theory introduced in the study, as its principal purpose is to broaden the application of canonical reactor theory. By discussing the relation between the various hyporheic zone river properties, the authors largely achieve this purpose, though I believe there are several concepts, clarfications, and theoretical advances (beyond Danckwerts) that would serve the reader if they were included in the discussion. I raise these points in the major comments section, as well as provide more specific comments, in the attached document.

Reviewer #3: The manuscript proposes the application of a series of concepts derived from chemical engineering to describe the main properties of hyporeic exchange in coarse-grained aquifers. Specifically, assuming a power law shape of the residence time distribution (RTD), it is discussed how hyporheic exchange can be summarized by two quantities, namely, the exponent of the power law (alpha) and the size of the hyporheic zone (s).

Even though the concepts expressed in the manuscript are not particularly novel (the fact that hyporheic exchange can be summarized by exchange flux and the RTD is in fact known), the way these concepts are summarized adapting an existing theory could represent a valid contribution. The limit of the manuscript is that the text could be more incisive, as the message is not well conveyed, if not unclear sometimes. More precisely, 1) the novel message should be better specified, 2) the application examples can be refined, and 3) some concepts should be clarified. All these issues are described in detail in the main comments below.

MAIN COMMENTS

1) NOVELTY SPECIFICATION: As I said, the manuscript does not present novel concepts (and this is correctly recognized in the text), so it is fundamental to state the key contribution of the work. In my view, the contribution is to choose a specific type of RTD (i.e., a one-parameter power law distribution) which is considered to be representative of hyporheic exchange, use this RTD to develop expressions for relevant quantities such as fluxes and age distributions (AD) as function of a few parameters (alpha and s), and suggest to employ this framework to classify streams by linking alpha and s to the characteristics of streams and catchments. If this was the intended aim of the manuscript, I recommend it to state more openly because it now gradually emerges from the text, and it is not evident to which extent the use of power law RTD is a mere example or a relevant assumption.

2) APPLICATION EXAMPLES: It is stated (line 359) that "chemical engineers have been applying these equations to bio-reactors (such as sewage treatment plants) to predict whole-system operation for decades [...]. Thus, a view of "streams as bioreactors" may provide a launchpad for potential collaborations between the engineering and ecological disciplines". It would then be extremely valuable to provide example of the applications. However, the applications of the theory that are suggested are not always very informative (e.g., temperature dynamics in fig 6; age of upwelling water in fig 4 works better), and I think that the reader is left out questioning how useful the proposed framework is. If a new approach is proposed, compelling examples should be provided. The same is true for Fig 7: most of the comments are about the limitations stemming from the assumptions, which is fine, but I would also stress the take-home message from the example.

3) CLARIFICATIONS: in a number of points, the text could be more straightforward and point out the aims of the concepts that are introduced, which are not always clear enough. Some examples follow:

- I think that the role of short residence times is overemphasized when it is said that "if we are willing to accept that hyporheic exchange scales according to a power law [...] our application of the equations assuming a power-law RTD reveals [that] the rate of hyporheic exchange in any given stream is dependent on the range of water ages considered" (lines 364-367). Starting from this point, almost a half of the discussion section deals with the implications of the power-law RTD; however, it is particularly the tails of RTDs that have been found to have power-law behavior, while this is not necessarily true for small times. In particular, the fractal behaviour of morphology (which leads to the fractal, nested system of hyporheic flow cells; line 387) should break up at small spatial scale due to physical constraints (e.g., at grain scale for topograpy-driven exchange, or at the Kolmogorov scale for turbulence-driven exchange). Moreover, at small scale different physical processes may prevail (i.e., diffusion rather than advection). What I am implying is that the role attributed to small residence times is likely to stop below some threshold time scale when the RTD may no longer be described by a power law. Because the manuscript is directed also to readers that are not familiar with hyporheic RTD, I think that the message "RTDs are always power-law distributed" could be misleading and the concepts expressed here may derive from the abnormal behaviour of the power-law distributions at small times when they could no longer be a good approximation of actual RT.

- The representative unit (RU) is defined at line 185 as "a conceptual unit of hyporheic water storage (s=1) which has, by definition, the same RTD and AD as the larger hyporheic zone it represents." This is probably not the best definition as it may lead to think that RUs are physical parts of the hyporheic zone (HZ), while it is unlikely that a portion of the hyporheic zone has the same RTD and AD than the total hyporheic zone. Later on it becomes clearer that the RU is introduced only to discuss the properties of HZ regardless of volume, i.e., an idealized HZ with unit volume. While the definition is formally correct, it does not make clear why the concept is introduced, leading to potential confusion for the reader.

- the manuscript "suggests that research efforts to identify stream characteristics correlated with values of alpha and s would expand opportunities for rapid characterization of hyporheic hydrology across stream networks [...], especially if such correlates were obtainable from readily available spatial data sets (digital elevation models, LIDAR, aerial photography, etc.)" (line 331-335). It is worth stressing the temporal variations of streamflow or groundwater flow also affect hyporheic exchange and should be factored when these data sets are built, or otherwise we would improperly attribute observed variations in hyporheic exchange to spatial rather than to temporal drivers. The picture is of course complex, but I think that it is important not to oversimplify it in this point.

- The notation is changed at line 221 by renaming many variables. Is this necessary? If this notation is simpler than the former one, why not using it throughout the manuscript?

- It is stated at line 197 that "For the remainder of this paper, we assume that the hyporheic RTD is proportional to a power law with a negative exponent" -> how critical is this assumption for the manuscript? See main comment 1.

OTHER COMMENTS

7 "expansive coarse-grained alluvial aquifers (ECGAAs)" -> I do not know what an expansive aquifer is. I recommend to explain it, even briefly.

73 it is said that q can represent, among other things, the "rate at which the cross-sectional area of hyporheic water exchanges with the cross-sectional area of the channel". This description is confusing (water flows through the cross section, and it is not straigthforward where it is exchanged betwenn channel and hyporheic zone). I suggest to describe it as the rate of water exchange per unit river length, which I think is a correct description.

122 "exit age density function" -> I think it is more coherent to refer it as residence time (RT) rather than age, since it has been already stated that RT denotes the time when water leaves the aquifer.

144 it would be useful to specify that W(t) is dimensionless, as the dimensions of other quantities have been reported.

153 missing "n" in "functioN"

177 this section is titled "Visualizing hydraulic geometry" and the nex one "Visualizing hyporheic hydraulic geometry". However, they both refer to the aquifer exchanging water with the channel, so the difference between the subject of the section is unclear. If there is no difference, titles should be changed.

181 "s can be approximated as the product of aquifer dimensions (length, width, and depth) and aquifer porosity." -> This is true only if s represent a volume, but as said before it can have different definitions. I suggest to better specify it here.

209 remove parentheses before and after "tau < 60 s".

Fig.3 I am not sure how informative this figure is.

240 "hyporehic" should be "hyporheic". Same at line 340.

250 "residence time" should be "water age"

311 "I(t) is not the same as, but can be derived from E(t)" -> is the other way around also true, as stated at line 445?

337 missing space after "hereafter"

340 "if more complex channels have higher rates of hyporehic exchange, such channels are likely to be associated with larger values of alpha." -> This is true only if the volume s is the same, while comparing river systems of different sizes could lead to very different results. I would rephrase slightly to avoid ambiguities.

355 "Considering multiple transient storage zones within the context of hyporheic hydraulic geometry provides the potential for a quantitatively rigorous but practical mechanism for scaling hyporheic heterogeneity to whole stream networks." -> the use of multiple transient storage zones is essentially a technical way to discretize water ages in finite classes. It is not very different than, e.g., using finite time steps in a particle tracking approach, or a discrete cells in a finite difference methods, and I would present it as such. Moreover, the link on how it can allow for upscaling is not well defined (see also main comment 2).

Reviewer #4: In this work the authors present a quantitative description of hydraulic geometry to visualize the interdependence among hydrologic variables such as age distribution and residence time distribution in the hyporheic zone. This is important because concepts related to AD and RTD are often misunderstood and misinterpreted and this paper will help to clarify and distinguish between these two variables. My own research pertains to hyporheic hydrology and I myself struggle with this distinction.

Overall, I very much enjoyed this paper and the rich development of equations describing the relationships between hyporheic flow, residence time, and storage. This paper is well-written, the theoretical constructs are well-developed and described, and the theme of the paper considers aspects of hyporheic zones that relate to scaling constructs that will be useful for non-modeling readers.

A main theme within this paper is to clarify differences and elucidate interdependencies between AD and the RTD. Lines 308-312 is one example where this distinction is highlighted and it has been noted elsewhere throughout the paper. Despite this prominent theme, after reading the paper, I still did not have a better understanding of how to conceptualize and understand the differences between the two. The authors are entrenched in the theoretical constructs and spend the majoring of the paper on this. Because of this, the conceptual development, and clear articulation of what these metrics actually mean in real life is lost. The temperature example did not help to resolve this issue for me. My suggestion is to resolve this using Figure 1. In Figure 1, you have already done the work showing TSZs in series, so can you provide a simple example of min/max ages for each TSZ within your conceptual model and associated residence times? And show how the pdf of the AD/RTD changes as a water particle moves through the TSZ. A simple example within your conceptual model would help distinguish between the two for the ecological/hydrological audience.

6. PLOS authors have the option to publish the peer review history of their article (what does this mean?). If published, this will include your full peer review and any attached files.

Reviewer #1: No

Reviewer #2: **Yes: **Kevin R Roche

Reviewer #3: No

Reviewer #4: No

---

## [Author Response · Author response to Decision Letter 0]

3 Sep 2021

We have attached a document entitled "Response_To_Reviewers.pdf" which provides in-depth responses to all reviewer comments.

---

## [Decision Letter · Decision Letter 1]

27 Oct 2021

PONE-D-21-04355R1Hyporheic hydraulic geometry: Ecological implications of relationships among hyporheic exchange, storage, and water agePLOS ONE

Dear Dr. Fogg,

Thank you for submitting your manuscript to PLOS ONE. After careful consideration, we feel that it has merit but does not fully meet PLOS ONE’s publication criteria as it currently stands. Therefore, we invite you to submit a revised version of the manuscript that addresses the points raised during the review process.

More specifically, I encourage you to take into consideration the concern of reviewer 3 on their second issue: lack of application examples. I wonder if another example could be provided to tackle this issue and make the manuscript less speculative.

We look forward to receiving your revised manuscript.

Kind regards,

Clara Mendoza-Lera

Academic Editor

PLOS ONE

Reviewers' comments:

Reviewer's Responses to Questions

**Comments to the Author**

1. If the authors have adequately addressed your comments raised in a previous round of review and you feel that this manuscript is now acceptable for publication, you may indicate that here to bypass the “Comments to the Author” section, enter your conflict of interest statement in the “Confidential to Editor” section, and submit your "Accept" recommendation.

Reviewer #3: (No Response)

2. Is the manuscript technically sound, and do the data support the conclusions?

Reviewer #3: Partly

3. Has the statistical analysis been performed appropriately and rigorously? 

Reviewer #3: N/A

4. Have the authors made all data underlying the findings in their manuscript fully available?

Reviewer #3: (No Response)

5. Is the manuscript presented in an intelligible fashion and written in standard English?

Reviewer #3: Yes

6. Review Comments to the Author

Reviewer #3: MAIN COMMENTS

In my previous review I identified three issues that were in need of attention: (1) insufficient novely specifications, (2) lack of application examples, and (3) need to clarify some paragraphs. The revisions made by the authors have substantially improved the manuscript in terms of issues (1) and (3): the introduction is now better structured and helps the reader in understanding the aim of the concepts later described in the manuscript, and many passages are now clearer.

In terms of issue (2), I still think that a manuscript that aims to provide a framework to help interpreting natural processes would greatly benefit from an example of application of this framework that is not purely theoretical. At present, the provided examples (fig. 4-7) are obtained from model simulations, and the illustrative example of water temperature is no different. I am not sure if the case study from which fig.2 was derived could provide such an applicative example, but in that case (or with any field data) the potential impact of the manuscript would greatly increase. Otherwise the manuscript is much more speculative, and I leave to the Editor the choice if the manuscript can be considered without this application.

OTHER COMMENTS (line numbers refer to the track-change manuscript)

6 "the top few hundredths of a meter" -> I would say "top centimeters", but this is a matter of taste

46 "rSAS" is firstly introduced here, and the acronym should hence be defined.

95 "a hydrosystems" -> "a hydrosystem"

Fig.6: the last sentence of the caption is missing something ("greater water ages and ."). Remove "and" or complete.

493 "While this problem may seem somewhat intractable" -> I think that my previous comment to this part was not completely clear: I understand the point the authors are raising, i.e., the method of analysis may lead to underestimating the flux because of the chosen scale. What I meant is that a low scale probably exist in most situation, because the RTD is unlikely to assume infinite values for small values of residence times, as the discussion here seems to imply. This is true for different types of RTD (e.g., for both power law and exponential distributions): at small residence time, the increase in probability may break up. I simply suggest to avoid giving the impression that the "true" flux can never be found even if a very fine scale of analysis is employed.

7. PLOS authors have the option to publish the peer review history of their article (what does this mean?). If published, this will include your full peer review and any attached files.

Reviewer #3: No

---

## [Author Response · Author response to Decision Letter 1]

15 Dec 2021

See response to reviewers document attached with submission.

---

## [Editor Report · Decision Letter 2]

19 Dec 2021

Hyporheic hydraulic geometry: Conceptualizing relationships among hyporheic exchange, storage, and water age

PONE-D-21-04355R2

Dear Dr. Fogg,

We’re pleased to inform you that your manuscript has been judged scientifically suitable for publication and will be formally accepted for publication once it meets all technical requirements.

Kind regards,

Clara Mendoza-Lera

Academic Editor

PLOS ONE
---

## [Editor Report · Acceptance letter]

30 Dec 2021

PONE-D-21-04355R2 

Hyporheic hydraulic geometry: Conceptualizing relationships among hyporheic exchange, storage, and water age. 

Dear Dr. Fogg:

I'm pleased to inform you that your manuscript has been deemed suitable for publication in PLOS ONE. Congratulations! Your manuscript is now with our production department. 

Kind regards, 

on behalf of

Dr. Clara Mendoza-Lera 

Academic Editor

PLOS ONE